# Radiomics Based on Thyroid Ultrasound Can Predict Distant Metastasis of Follicular Thyroid Carcinoma

**DOI:** 10.3390/jcm9072156

**Published:** 2020-07-08

**Authors:** Mi-ri Kwon, Jung Hee Shin, Hyunjin Park, Hwanho Cho, Eunjin Kim, Soo Yeon Hahn

**Affiliations:** 1Department of Radiology, Kangbuk Samsung Hospital, Sungkyunkwan University School of Medicine, Seoul 03181, Korea; kwonmrrad@gmail.com; 2Department of Radiology, Samsung Medical Center, Sungkyunkwan University School of Medicine, Seoul 06351, Korea; aurore47@naver.com; 3School of Electronic and Electrical Engineering, Sungkyunkwan University, Jangan-gu, Suwon 16419, Korea; 4Center for Neuroscience Imaging Research, Institute for Basic Science, Jangan-gu, Suwon 16419, Korea; 5Department of Electrical and Computer Engineering, Sungkyunkwan University, Jangan-gu, Suwon 16419, Korea; nara9313@gmail.com (H.C.); dmswlskim970606@gmail.com (E.K.)

**Keywords:** radiomics, follicular thyroid carcinoma, distant metastasis, ultrasonography, support vector machine

## Abstract

We aimed to evaluate whether radiomics analysis based on gray-scale ultrasound (US) can predict distant metastasis of follicular thyroid cancer (FTC). We retrospectively included 35 consecutive FTCs with distant metastases and 134 FTCs without distant metastasis. We extracted a total of 60 radiomics features derived from the first order, shape, gray-level cooccurrence matrix, and gray-level size zone matrix features using US imaging. A radiomics signature was generated using the least absolute shrinkage and selection operator and was used to train a support vector machine (SVM) classifier in five-fold cross-validation. The SVM classifier showed an area under the curve (AUC) of 0.90 on average on the test folds. Age, size, widely invasive histology, extrathyroidal extension, lymph node metastases on pathology, nodule-in-nodule appearance, marked hypoechogenicity, and rim calcification on the US were significantly more frequent among FTCs with distant metastasis compared to those without metastasis (*p* < 0.05). Radiomics signature and widely invasive histology were significantly associated with distant metastasis on multivariate analysis (*p* < 0.01 and *p* = 0.003). The classifier using the results of the multivariate analysis showed an AUC of 0.93. The radiomics signature from thyroid ultrasound is an independent biomarker for noninvasively predicting distant metastasis of FTC.

## 1. Introduction

Follicular thyroid carcinoma (FTC) is the second most common thyroid cancer and constitutes approximately 10–15% of all thyroid malignancies [1]. It has an indolent disease course and good clinical outcome, except in cases of distant metastasis. Distant metastasis via hematogenous spread occurs in 5.1–11% of patients with FTC, most commonly involving the lung and bone, and is a significant independent prognostic factor for poor survival of patients with FTC [2,3,4,5].

Distant metastasis may be the initial presentation of the disease or it may occur after the initial treatment for cancer. Even if distant metastasis is not clinically evident at the time of diagnosis, early recognition and aggressive treatment of FTCs with high risk of distant metastasis would be invaluable in planning an appropriate treatment strategy. Old age, invasive histological classification, large tumor size (>4 cm), extensive vascular invasion, and lymph node metastasis are independent risk factors for distant metastasis of FTCs [3,5,6,7,8]. One study by Kim et al. [9] found that preoperative ultrasound (US) features as well as clinicopathologic characteristics can predict distant metastasis. US features including marked hypoechogenicity, the presence of a nodule-in-nodule appearance, rim calcification, age, and invasive histology were significantly associated with distant metastasis.

Radiomics, which extracts high dimensional features from medical images, has recently emerged and received great attention as an innovative tool in cancer research [10,11]. Researchers can mine crucial information that is difficult to discern with the naked eye but reflects the underlying pathophysiology that could potentially provide information about cancer prognosis as well as diagnosis. Biomarkers based on quantitative radiomics features, commonly referred to as the radiomics signature, have shown promising results for decision support in various types of cancers, including thyroid cancer [12,13,14,15,16].

Knowing which predictors are associated with distant metastasis in patients diagnosed with FTC may influence preoperative planning, may prevent an unnecessary second operation, and may be useful in determining the need for radioiodine therapy. To our knowledge, there have been no published studies aimed at predicting distant metastasis in FTC based on radiomics analysis. We hypothesized that the substantial objective information extracted from US images, distinguishable through visual interpretation, can predict distant metastasis in FTC. Moreover, if a noninvasive, objective method such as radiomics is effective, it may be possible to achieve personalized management strategies.

The purpose of our study was to determine whether a radiomics signature based on preoperative gray-scale US could predict distant metastasis in FTC.

## 2. Materials and Methods

### 2.1. Patient Selection and Data Collection

Our institutional review board approved this retrospective study. The informed consent was waived. We retrospectively searched our institutional database to identify patients with surgically confirmed FTC who underwent preoperative thyroid US between January 1998 and December 2016 and had follow-up surveillance data. The exclusion criteria were as follows: (1) nodule diameter <10 mm, as the region of interest (ROI) method has lower accuracy and current guidelines do not recommend fine needle aspiration (FNA) for nodules with a diameter <10 mm [17,18]; (2) lack of precise correlation between pathology and US findings in patients with multiple nodules; and (3) poor image quality, preventing precise nodule discrimination. There was a total of 35 consecutive FTCs with distant metastasis. We included 134 consecutive FTCs without distant metastasis diagnosed between January 2012 to December 2016 as the control group. Distant metastasis was confirmed by biopsy and/or diagnosed by imaging modalities including computed tomography (CT), positron emission tomography (PET), or magnetic resonance imaging (MRI).

We extracted clinical characteristics from electronic medical records including age at diagnosis, sex, type of surgery (i.e., hemithyroidectomy or total thyroidectomy), and the time of diagnosis of distant metastasis. We also retrospectively collected pathologic findings based on final pathologic reports after surgery including tumor size, histologic classification according to the extent of invasiveness (minimally invasive or widely invasive), the extrathyroidal extension (none, minimal, (extension to the sternothyroid muscle or perithyroidal soft tissue), or gross (extension to the subcutaneous soft tissue, larynx, trachea, esophagus, or recurrent laryngeal nerve)), and lymph node metastasis.

### 2.2. US Examinations and Image Evaluation

All patients underwent preoperative thyroid US using a HDI5000 (Advanced Technology Laboratories, Bothell, WA, USA), Logic 700 (GE Healthcare, Milwaukee, WI, USA) or an iU22 system (Vision 2010; Philips, Seattle, WA, USA) with a commercially available 5- to 12-MHz linear-array transducer before surgery. All scans were performed by one of seven radiologists with 2 to 17 years of experience in thyroid US.

Two radiologists (M.R.K. and J.H.S with 3 and 17 years of experience in thyroid US, respectively) retrospectively reviewed preoperative US and assessed image features. Radiologists were blinded to the presence of distant metastasis and clinicopathologic variables when reviewing US. Discordant cases were discussed to reach a consensus. According to the Korean thyroid imaging reporting and data system (K-TIRADS), all thyroid nodules were evaluated for internal content, echogenicity, shape, orientation, margin, calcification, presence of halo, and vascularity [17]. The presence of a nodule-in-nodule appearance, which appears as a conglomeration of small nodules within a well-defined solid nodule, was evaluated [9,19]. Nodules were classified according to the K-TIRADS categories as follows: category 1, no nodule; category 2, benign nodule; category 3, low suspicion nodule; category 4, intermediate suspicion nodule; and category 5, high suspicion nodule. The detection of suspicious lymph nodes on preoperative US was recorded. Cystic change, calcification, hyperechogenicity, and abnormal vascularity were considered suspicious features of cervical lymph nodes [17].

### 2.3. Radiomics Feature Analysis

A study process is summarized in Figure 1. A single radiologist (J.H.S) selected the most representative image among US images of each tumor for radiomics feature extraction. An ROI was manually delineated along the border of each tumor on representative US images using the Microsoft Paint program (Microsoft Corporation, Redmond, WA, USA) by two radiologists (M.R.K and J.H.S.). Reproducibility in radiomics features is an important topic and intraclass correlation coefficient (ICC) was computed to assess the reproducibility of features using two sets of ROIs. The first set of ROIs was used for the radiomics analysis.

A total of 60 radiomics features were extracted using an open-source radiomics software, PyRadiomics. The features were grouped into first order (17 features), 2D shape (3 features), gray-level cooccurrence matrix (GLCM, 24 features), and gray-level size zone matrix features (GLSZM, 16 features). The histogram-based features were computed from 32-bin histograms calculated over the intra-tumoral intensity range. The GLCM features assess textural information and reflect intra-tumoral heterogeneity. They were computed using a 2D histogram with 32 bins. A total of eight matrices corresponding to eight 2D directions with an offset of one were computed and then averaged to yield a single matrix. The averaged matrix was used to compute the GLCM features. The GLSZM features are also texture features assuming that a tumor consists of many gray-level zones with different sizes. They were computed using a 16 × 256 matrix, where the first dimension was binned gray level and the second dimension was the size of the gray level zone. Further details of the radiomics features are publicly available.

Due to the lack of external validation data, we applied five-fold cross-validation to separate our data into training and test sets to reduce the risk of overfitting. Models were built using the training set only and tested on a left-out test set. Feature selection was performed using the least absolute shrinkage and selection operator (LASSO) method from the training set. The selected features were used as inputs to train a support vector machine (SVM) using the linear kernel. The trained classifier was further tested on a left-out test fold. A radiomics signature was computed as the probability of the SVM classifier output for each nodule from the test-fold. In detail, the SVM classifier with a linear kernel was constructed using the selected features. The parameters of SVM were weights that determined the decision hyperplane to separate two groups (i.e., FTC with and without metastasis). A signed distance was computed from features from the given sample to the hyperplane, which was further transformed using the sigmoid function to yield a probability value. The output probability value was assigned as the radiomics signature. The whole procedure was performed with a MATLAB command “fticsvm” with the prior uniform option. As we adopted five-fold cross-validation, we repeated the feature selection, model training, and testing steps five times, each leaving out a different test fold. The performance of the classifier models was assessed based on the area under the receiver operating characteristic curve (AUC), accuracy, sensitivity, and specificity. The software code for the whole procedure is given in the Appendix A.

### 2.4. Statistical Analysis and Combined Model

FTCs with and without distant metastasis were compared based on patient demographic information. Categorical variables were analyzed using the Chi-square test, while continuous variables were analyzed using Student’s *t*-test.

Univariate analysis was performed to evaluate the association between distant metastases of FTC and radiomics signature, US findings, and clinicopathologic variables. The correlation between continuous variables was assessed with Pearson’s correlation analysis, that between binary variables was assessed with Matthew’s correlation analysis, and that between binary and continuous variables was assessed with point biserial correlation analysis. The univariate association of radiomics signature was evaluated using the same five-fold cross-validation with averaged correlation and *p*-values. Multivariate logistic regression analysis was performed using variables that showed statistically significant differences on univariate analysis to identify independent predictors of distant metastasis of FTC using the same five-fold cross-validation. A *p*-value < 0.05 was considered statistically significant. A final SVM classifier was built using significant variables from the multivariate analysis, and the performance was assessed using the same five-fold cross-validation.

## 3. Results

The clinicopathologic characteristics of the study population are shown in Table 1. Among the total of 169 patients, 78 were women (mean age: 45.1 ± 13.9 years; range: 19–77 years) and 18 were men (mean age: 43.9 ± 10.2 years; range: 21–62 years). Hemithyroidectomy was performed in 81 patients (48.9%), while total thyroidectomy was performed in 88 patients (52.1%).

Of the 35 patients with distant metastases, the initial presentation of 24 patients was distant metastases and follicular cancers were diagnosed later. The remaining 11 patients developed distant metastasis during follow-up. Three patients were found to have metastases within six months after surgery. Eight patients developed distant metastasis one to three years after surgery. The sites of distant metastasis were bone alone (*n* = 14), followed by both bone and lung (*n* = 12), lung alone (*n* = 7), and multiple sites (*n* = 2), that is bone, lung, and brain (*n* = 1) and bone, lung, and kidney (*n* = 1). Patients with distant metastasis were older than those without metastasis (*p <* 0.0001). The mean pathologic tumor size was larger in FTCs with distant metastasis than without metastasis (*p* = 0.033). Widely invasive histology, lymph node metastasis, and extrathyroidal extension were more frequent in patients with FTC who had distant metastasis than in those who did not (*p* < 0.01). Of the 134 patients without distant metastasis, a median follow-up period was 5.7 years (range: 3.3–8.3 years).

The US characteristics of the study population are presented in Table 2. Of the preoperative US features of the thyroid nodules, marked hypoechogenicity, rim calcification, nodule-in-nodule appearance, and the presence of suspicious lymph nodes on US were more frequent in FTCs with distant metastasis than in those without metastasis (*p* < 0.05). FTCs with distant metastasis had higher incidences of K-TIRADS category 5 (high suspicion) and 4 (intermediate suspicion) compared to FTCs without distant metastasis (*p* < 0.0001).

We used five-fold cross-validation, and thus, the selected features varied from fold-to-fold. Six features were selected in all five training folds. They were the minimum (first order), elongation and sphericity (shape), gray-level nonuniformity normalized, size zone nonuniformity, and small area low gray-level emphasis (GLSZM). The SVM classifier showed a high AUC of 0.90 on average on the test folds. The performance of SVM classifiers is shown in Table 3.

At univariate analysis, the radiomics signature was associated with distant metastasis (*p* < 0.001) (Table 4) and there was a significant difference (*p* < 0.0001) in radiomics signature between two groups (i.e., FTC with and without metastasis). Among the clinicopathologic variables, age, size, widely invasive histology, lymph node metastasis, and extrathyroidal extension were associated with distant metastasis (*p* < 0.05). Among the US characteristics, orientation, echogenicity, rim calcification, and nodule-in-nodule appearance were associated with distant metastasis (*p* < 0.05). At multivariate analysis, only radiomics signature and widely invasive histology were independently associated with distant metastasis (*p* < 0.01 and *p* = 0.003, respectively) (Table 4). 

Diagnostic performance of predictors using the results of multivariate analysis (radiomics signature and widely invasive histology) for predicting metastasis in FTC patients showed a high AUC of 0.93 on the test folds (Table 3).

The ICC for the six important features were 0.99, 0.96, 0.81, 0.82, 0.95, and 0.96 for the minimum (first order), elongation and sphericity (shape), gray level non-uniformity normalized, size zone nonuniformity, and small area low gray-level emphasis (GLSZM) features, respectively, showing excellent agreement. The ICC for all 60 radiomics features was 0.94 on average. The full list of ICC values is given in Appendix A.

The selected radiomics features could be correlated with other important clinical variables. Thus, we correlated six important radiomics features common in all five folds with the important clinical variables that were available before surgery (i.e., tumor size, nodule-in-nodule appearance, rim calcification, and echogenicity) [9]. The correlation analyses showed that the selected radiomics features had either weak correlation (*r* < 0.2) or high *p*-value (*p* > 0.1) (Appendix A). This confirmed that the selected radiomics features were not surrogates of other important clinical variables.

## 4. Discussion

Since the introduction of radiomics, many studies concluded that it has potential power for analyzing thyroid cancers, including the differentiation between malignant and benign nodules or the prediction of prognosis or gene mutations in papillary thyroid carcinoma [12,20,21,22,23,24,25]. However, little is known about the association of radiomics and FTC. To our knowledge, this is the first study to apply radiomics in predicting distant metastasis of FTC. In the current study, we evaluated whether radiomics, clinicopathologic, and US characteristics could be independent predictors of distant metastasis in FTC. Old age, larger tumor size, widely invasive histology, extrathyroidal extension, and lymph node metastases were found more frequently in FTCs with distant metastasis, consistent with previous studies [3,5,6,7,8]. Visual assessment of thyroid nodules on preoperative US suggested that suspicious US features, including marked hypoechogenicity, nodule-in-nodule appearance, rim calcification, and the presence of suspicious lymph nodes, were also more frequent in FTCs with distant metastasis. However, US features were not significantly associated with distant metastasis on multivariate analysis.

At multivariate analysis, only widely invasive histology and radiomics signature were independent risk factors for distant metastasis in FTC, and these predictors achieved a high AUC value of 0.93 on the test folds. Widely invasive histology is classified as high-risk by the American Thyroid Association guidelines, which supports our results [26]. As a high-risk cancer, widely invasive histology requires total thyroidectomy and is followed by radioiodine ablation therapy and/or cross-sectional imaging. However, invasive histologic findings are confirmed on pathological examination only after diagnostic surgery. In addition, it is impossible to preoperatively differentiate between minimally and widely invasive FTC as well as between FTC and benign follicular adenoma based on cytology, as follicular cell-derived thyroid nodules share overlapping cytomorphological characteristics [27]. We examined the performance of the classifier model using only radiomics features, and this also achieved a high AUC value of 0.90 on the test folds. This result suggests that radiomics can help optimize preoperative management in an essentially personalized manner.

In fact, radiomics is usually performed using tomographic images, including CT, PET, or MRI, because these modalities can acquire 3D volume data and data acquisition can be standardized by setting scan parameters identically [10]. Contrary to tomographic images, US has several limitations in quantitative analysis including uncommon 3D data available, lack of reproducibility, lack of representative features due to limited amount of image data, operator dependency, and dependency on US machines. However, US is the most widely used standard imaging tool for evaluating thyroid pathology and many recent studies concluded that quantitative features extracted from thyroid US images have favorable results [12,20,21,22,23,24,25].

Our results imply that radiomics goes beyond visual inspection of image. Although several suspicious US features were more common in FTCs with distant metastasis, they failed to show association with metastasis in multivariate analysis, but the radiomics signature remained as an independent marker for predicting distant metastasis. Our radiomics signature was derived from the selected features in the test fold within the cross-validation framework. That led to having a different number of features selected in each fold. However, six features were common in all five folds and many of them have interpretable implications that might lead to explaining the potential mechanism. The two shape features of elongation and sphericity were selected, and they imply that less elongated and anti-spherical nodules tend to be more malignant and thus could be metastatic based on widely accepted thyroid malignant US features [17]. The texture features from GLSZM (i.e., gray-level nonuniformity normalized, size zone nonuniformity, and small area low gray-level emphasis) all quantify intra-nodular heterogeneity in intensity, which plays an important in metastasis in many cancers [28,29]. The key to identifying the presence of metastasis in FTC in our study was that FTC with metastasis has obviously aggressive features that could not be explained through US descriptors only compared to FTC without metastasis.

Machine learning classifiers require sufficient samples to be trained properly. We used a different set of features (on average 10 features across different folds) from 169 samples (134 FTC without metastasis and 35 FTC with metastatic cases) for the SVM classifier. A theoretic study pointed out that, if the features followed multivariate Gaussian distribution, the number of samples per class to apply SVM effectively should be greater than three times the number of features [30]. In our case, the cutoff was 30 (=10 × 3) on average due to the different number of selected features in the cross-validation. Thus, we have a theoretic rationale to apply SVM to our samples.

There are several limitations to our study. First, this study was retrospective using data from a single institution. FTCs with distant metastasis are relatively rare in comparison to FTCs without metastasis. For statistical analysis, the enrollment periods between two groups were different but consecutive periods were used. This is a natural limitation of the disease. Second, using different US machines during the study period could have introduced machine-dependent bias. One study reported that their radiomics models were dependent upon the type of US machine due to innate differences in features such as brightness or contrast [31]. Cases in our study have been collected for a long period of time, but 96.4% of cases was executed with the most recent version. Differences in radiomics features due to machine changes appear to be small. Another limitation is the lack of external validation. This is difficult due to the low prevalence of FTCs with distant metastasis. Finally, the median follow-up period (5.7 years) for patients without distant metastasis was relatively short, and there is a possibility of underestimation of delayed distant metastasis. According to previous reports, delayed distant metastasis developed at a median of 4.5 years after surgery [32]. However, more than half of our patients (61.2%, 82/134) had a follow-up period of over 5 years.

In conclusion, the radiomics signature from thyroid ultrasound is an independent biomarker for noninvasively predicting distant metastasis in FTC preoperatively.

## Figures and Tables

**Figure 1 jcm-09-02156-f001:**
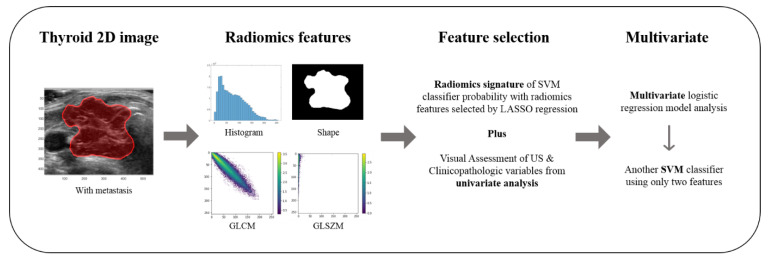
The study process: The most representative image of each tumor on thyroid 2D ultrasound (US) image was selected. Radiomics features including first order, shape, gray-level cooccurrence matrix (GLCM), and gray-level size zone matrix features (GLSZM) were extracted. A radiomics signature was generated using the least absolute shrinkage and selection operator (LASSO) and was used to train a support vector machine (SVM) classifier in five-fold cross-validation. For determining association between the radiomics signature, US findings, clinicopathological variables, and distant metastasis, univariate and multivariate logistic regression analyses were performed. Another SVM classifier was built using significant variables from the multivariate analysis.

**Table 1 jcm-09-02156-t001:** Clinicopathologic characteristics of the 169 patients with follicular thyroid carcinoma with and without distant metastasis.

Characteristics	Without Metastasis (*n* = 134)	With Metastasis (*n* = 35)	*p*-Value
Age (years)	46.51 ± 14.09	59.35 ± 12.0	<0.0001
Sex			0.214
Female	99 (73.9)	30 (85.7)	
Male	35 (26.1)	5 (14.3)	
Size	3.25 ± 1.52	4.03 ± 2.86	0.033
Histology			<0.0001
Minimally invasive	129 (96.3)	15 (42.9)	
Widely invasive	5 (3.7)	20 (57.1)	
Extrathyroidal extension			<0.0001
None	128 (95.52)	23 (65.71)	
Minimal	6 (4.48)	8 (22.86)	
Gross	0 (0)	4 (11.43)	
Lymph node metastasis			0.006
Absence	133 (99.25)	31 (88.57)	
Presence	1 (0.75)	4 (11.43)	

Numeric data are presented as mean ± standard deviation. Nonnumeric data are presented as the number of patients (percentage).

**Table 2 jcm-09-02156-t002:** Ultrasonographic (US) features of the 169 follicular thyroid carcinomas with and without distant metastasis.

US Features	Without Metastasis (*n* = 134)	With Metastasis (*n* = 35)	*p*-Value
Internal content			0.221
Solid	89 (66.42)	28 (80.0)	
Predominantly solid	44 (32.84)	6 (17.14)	
Predominantly cystic	1 (0.75)	1 (2.86)	
Echogenicity			<0.0001
Marked hypoechogenicity	18 (13.43)	16 (45.71)	
Hypoechogenicity	49 (36.57)	12 (34.29)	
Isoechogenicity	66 (49.25)	7 (20.0)	
Hyperechogenicity	1 (0.75)	0 (0)	
Shape			0.710
Irregular	16 (11.94)	5 (14.29)	
Oval to round	118 (88.06)	30 (85.71)	
Orientation			0.931
Nonparallel	8 (5.97)	6 (17.14)	
Parallel	126 (94.03)	29 (82.86)	
Margin			0.183
Spiculated/microlobulated	10 (7.46)	6 (17.14)	
Ill-defined	8 (5.97)	1 (2.86)	
Smooth	116 (86.57)	28 (80.8)	
Calcification			<0.0001
No	96 (71.64)	6 (17.14)	
Microcalcification	4 (2.99)	2 (5.71)	
Macrocalcification	12 (8.96)	3 (8.57)	
Rim calcification	22 (16.42)	24 (68.57)	
Halo			0.308
No	22 (16.42)	9 (25.71)	
Yes	112 (83.58)	26 (74.29)	
Vascularity			0.206
No	4 (2.99)	3 (8.57)	
Perinodular	18 (13.43)	2 (5.71)	
Mild	36 (36.87)	6 (17.14)	
Marked	29 (21.64)	7 (20.0)	
Not done	47 (35.07)	17 (48.57)	
Nodule-in-nodule appearance			<0.0001
No	110 (82.09)	13 (37.14)	
Yes	24 (17.91)	22 (62.86)	
K-TIRADS category			<0.0001
3 (low suspicion)	76 (56.72)	7 (20.00)	
4 (intermediate suspicion)	48 (35.82)	18 (51.43)	
5 (high suspicion)	10 (7.46)	10 (28.57)	
Suspicious lymph node			0.037
Presence	1 (0.75)	3 (8.57)	
Absence	133 (99.25)	32 (91.43)	

Data are presented as the number of patients (percentage). K-TIRADS: Korean Thyroid Imaging Reporting and Data System.

**Table 3 jcm-09-02156-t003:** Classification results and diagnostic performance of the predicting factors for distant metastasis in FTC patients using support vector machine during five-fold cross-validation.

Predictors		Radiomics Features	Radiomics Signature and Widely Invasive Histology
Training performance	AUC	0.93	0.94
	Accuracy	0.88	0.88
	Sensitivity	0.92	0.94
	Specificity	0.87	0.86
Test performance	AUC	0.90	0.93
	Accuracy	0.85	0.87
	Sensitivity	0.80	0.91
	Specificity	0.87	0.86

AUC: area under the receiver operating characteristic curve.

**Table 4 jcm-09-02156-t004:** Univariate correlation and multivariate logistic regression analysis associated with distant metastases of FTC.

Variables	Univariate		Multivariate (5-Fold Averaged)	
	Correlation Coefficient	*p*-Value	Beta Coefficient	*p*-Value
Age	0.356	<0.0001	0.022	0.502
Tumor size	0.165	0.033	−0.024	0.431
Widely invasive histology	0.610	<0.0001	0.335	0.003
Extrathyroidal extension	0.417	<0.0001	0.039	0.349
Lymph node metastasis	0.256	0.001	−0.033	0.684
Nodule-in-nodule appearance	0.409	<0.0001	0.069	0.361
Echogenicity	−0.330	<0.0001	−0.003	0.649
Orientation	−0.164	0.033	0.088	0.446
No calcification	−0.452	<0.0001	−0.109	0.221
Rim calcification	0.475	<0.0001	0.039	0.478
K-TIRADS category	0.342	<0.0001	−0.029	0.541
Suspicious lymph node on US	0.209	0.007	0.037	0.880
Radiomics signature *	0.649	<0.0001	0.079	<0.0001

* The result of radiomics signature in univariate analysis is an averaged value from 5-fold cross-validation.

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
