# Peer review of "Radiomics Based on Thyroid Ultrasound Can Predict Distant Metastasis of Follicular Thyroid Carcinoma"

_jcm, 2020, doi:10.3390/jcm9072156_

Round 1

Reviewer 1 Report

In this study, the authors evaluated whether radiomics analysis based on a gray-scale ultrasound can predict distant metastasis of follicular thyroid cancer (FTC).  They found that radiomics signature and widely invasive histology were significantly associated with distant metastasis and concluded that radiomics signature to be an independent biomarker useful for predicting metastasis of FTC. The manuscript is well written, and the conclusion sounds reasonable.

Minor points: 

  1. Methods. How was distant metastasis confirmed?
  2. Results. Some numerics appear both in sentences in Result and Tables. For example, ”(with metastasis: 59.35 ± 12.0 years, without 171 metastasis: 46.51 ± 14.09 years, p < 0.0001” appears in Table 1.
  3. Discussion, line 238. There are some studies on 3D-US.
  4. Discussion, line 243-. I understand that the radiomics signature was a significant factor that predicted distant metastasis. What do the authors think to be the potential mechanism that caused these results? Is it a black box?

Author Response

Review 1

Comments and Suggestions for Authors

In this study, the authors evaluated whether radiomics analysis based on a gray-scale ultrasound can predict distant metastasis of follicular thyroid cancer (FTC).  They found that radiomics signature and widely invasive histology were significantly associated with distant metastasis and concluded that radiomics signature to be an independent biomarker useful for predicting metastasis of FTC. The manuscript is well written, and the conclusion sounds reasonable.

Minor points: 

  1. How was distant metastasis confirmed?

>> Thank you for your question. Distant metastasis was confirmed by biopsy and/or diagnosed by imaging modalities including computed tomography, positron emission tomography or magnetic resonance imaging. We added it in the Material and methods.

  1. Some numerics appear both in sentences in Result and Tables. For example, ”(with metastasis: 59.35 ± 12.0 years, without 171 metastasis: 46.51 ± 14.09 years, p < 0.0001” appears in Table 1.

>> Thank you for your comment. To avoid duplication, we removed duplicated numerics appearing both in the manuscript and tables.

  1. Discussion, line 238. There are some studies on 3D-US.

>> Thank you for your comment. I want to emphasize that ultrasound usually obtains 2D data. Since the introduction of 3D-US, it has not yet been widely used in clinical practice yet, especially in the evaluation of thyroid nodules.

  1. Discussion, line 243-. I understand that the radiomics signature was a significant factor that predicted distant metastasis. What do the authors think to be the potential mechanism that caused these results? Is it a black box?

>> Six features were common in all five folds and many of them have interpretable implications that might lead to explaining the potential mechanism. Two shape features of elongation and sphericity were selected and they imply that less elongated and anti-spherical nodules tend to be more malignant and thus could be metastatic, based on widely accepted thyroid malignant US features [17]. The texture features from GLSZM (i.e., gray level non-uniformity normalized, size zone nonuniformity, and small area low gray-level emphasis) all quantify intra-nodular heterogeneity in intensity, which plays an important in metastasis in many cancers [28,29].

Reviewer 2 Report

This study investigates the role of ultrasound radiomics in predicting distant metastasis of the follicular thyroid carcinoma (FTC). To that aim, the authors enrolled 169 patients (35 of which had distant metastasis), and extracted 60 radiomics features (RF) from each of 2D US images. Since no validation set was available, 5-fold cross-validation technique was performed, leaving each time a different sample as validation set. LASSO algorithm was used to select the non-redundant features: the RFs selected in all five training folds were used to build the radiomics signature. The predictive power of radiomics score, clinical data and US information were evaluated with univariate correlation and multivariate logistic regression analyses. Finally, two support vector machine classifiers were trained using only the radiomics score and all the statistically significant variables, respectively. The authors found that a six radiomics features score and widely invasive histology were independently associated with distant metastasis providing good predictive performance.

General Comment:

The study is of interest, however it is affected by some limitations as underlined also by the authors:

  1. The enrollment time is not the same for patients with and without metastases. This also leads to include images acquired with different ultrasound system technologies, which may provide not reliable results.
  2. The US imaging is one of the most operator dependent technique and the RFs are highly affected by acquisition settings; therefore, the agreement between RFs extracted from images acquired by different operators is mandatory.
  3. It was demonstrated that RFs could be correlated with others clinical parameters; since authors did not performed any correlation analysis, it cannot be excluded that radiomics score was a surrogate of other clinical variables.
  4. The use of SVM classifier for this sample size is not justified and the risk of overfitting is very high.
  5. The authors did not provide any information about the calculated radiomics score as well as the full details of predictive model, and this does not allow to replicate the study.

Specific Comment:

Line 74: Why authors decided to reduce the enrollment time for patients without distant metastases?

Line 88: Authors should compare the radiomics features extracted from images acquired by the different operators.

Line 148: Usually univariate logistic regression is performed before the multivariate one: why did the authors do a correlation analysis?

Line 186: Is the radiomics score a linear combination of the six LASSO selected features? Is it statistically different between the two groups of patients?

Line 203: Authors should report the ICC results transparently.

Line 205: Usually, the model parameters has to be reported in order to make the analysis repeatable.

Line 254: The authors stated that only 6/169 patients were acquired with an old ultrasound system: why did they not exclude these patients from the analysis?

Final suggestion

Although the study is of interest, some important limitations affect this research; therefore, I do not recommend the manuscript for publication.

Author Response

Comments and Suggestions for Authors

This study investigates the role of ultrasound radiomics in predicting distant metastasis of the follicular thyroid carcinoma (FTC). To that aim, the authors enrolled 169 patients (35 of which had distant metastasis), and extracted 60 radiomics features (RF) from each of 2D US images. Since no validation set was available, 5-fold cross-validation technique was performed, leaving each time a different sample as validation set. LASSO algorithm was used to select the non-redundant features: the RFs selected in all five training folds were used to build the radiomics signature. The predictive power of radiomics score, clinical data and US information were evaluated with univariate correlation and multivariate logistic regression analyses. Finally, two support vector machine classifiers were trained using only the radiomics score and all the statistically significant variables, respectively. The authors found that a six radiomics features score and widely invasive histology were independently associated with distant metastasis providing good predictive performance.

General Comment:

The study is of interest, however it is affected by some limitations as underlined also by the authors:

  1. The enrollment time is not the same for patients with and without metastases. This also leads to include images acquired with different ultrasound system technologies, which may provide not reliable results.

>> Thank you for your comment. Generally, the number of FTCs with distant metastasis was much smaller than FTCs without distant metastasis. For statistical analysis, FTC with distant metastasis included all data from our institution and FTC without metastasis was limited to a period of four consecutive years.

  1. The US imaging is one of the most operator dependent technique and the RFs are highly affected by acquisition settings; therefore, the agreement between RFs extracted from images acquired by different operators is mandatory.

>> We agree with you! The agreement in radiomics features is an important issue, which was evaluated with ICC. However, the previous description of ICC was insufficient, thus we expanded the section on ICC. A full list of ICC values for all 60 radiomics features are reported in the Supplement as a Table S1 (repeated below for your review). The ICC values of the six selected features were mentioned in the main text. We expanded the Methods and Results sections.

Supplemenatary Table S1. Intraclass coefficient of each radiomics feature

Radiomics features

(n=60)

ICC

Radiomics features

(n=60)

ICC

Firstorder_90Percentile

0.9850

GLCM_IDM

0.9379

Firstorder_Energy

0.9822

GLCM_IDMN

0.9173

Firstorder_Entropy

0.9036

GLCM_IDN

0.9295

Firstorder_InterquartileRange

0.9704

GLCM_IMC1

0.9691

Firstorder_Kurtosis

0.9504

GLCM_IMC2

0.9741

Firstorder_Maximum

0.8838

GLCM_InverseVariance

0.9755

Firstorder_MeanAbsoluteDeviation

0.9776

GLCM_JointAverage

0.9259

Firstorder_Mean

0.9912

GLCM_JointEnergy

0.9737

Firstorder_Median

0.9933

GLCM_JointEntropy

0.9152

Firstorder_Minimum

0.9939

GLCM_MCC

0.9755

Firstorder_Range

0.8792

GLCM_MaximumProbability

0.9736

Firstorder_RobustMeanAbsoluteDeviation

0.9771

GLCM_SumAverage

0.9259

Firstorder_RootMeanSquared

0.9888

GLCM_SumEntropy

0.9116

Firstorder_Skewness

0.9532

GLCM_SumSquares

0.8957

Firstorder_TotalEnergy

0.9822

GLSZM_GrayLevelNonUniformity

0.9732

Firstorder_Uniformity

0.9407

GLSZM_GrayLevelNonUniformityNormalized

0.8215

Firstorder_Variance

0.9705

GLSZM_GrayLevelVariance

0.8788

Shape_Elongation

0.9572

GLSZM_HighGrayLevelZoneEmphasis

0.8554

Shape_PerimeterSurfaceRatio

0.9746

GLSZM_LargeAreaEmphasis

0.9603

Shape_Sphericity

0.8062

GLSZM_LargeAreaHighGrayLevelEmphasis

0.8366

GLCM_Autocorrelation

0.8912

GLSZM_LargeAreaLowGrayLevelEmphasis

0.9889

GLCM_ClusterProminence

0.8465

GLSZM_LowGrayLevelZoneEmphasis

0.9625

GLCM_ClusterShade

0.9272

GLSZM_SizeZoneNonUniformity

0.9523

GLCM_ClusterTendency

0.8966

GLSZM_SizeZoneNonUniformityNormalized

0.9593

GLCM_Contrast

0.9174

GLSZM_SmallAreaEmphasis

0.9403

GLCM_Correlation

0.9759

GLSZM_SmallAreaHighGrayLevelEmphasis

0.8480

GLCM_DifferenceAverage

0.9285

GLSZM_SmallAreaLowGrayLevelEmphasis

0.9614

GLCM_DifferenceEntropy

0.9255

GLSZM_ZoneEntropy

0.9336

GLCM_DifferenceVariance

0.9148

GLSZM_ZonePercentage

0.9210

GLCM_ID

0.9411

GLSZM_ZoneVariance

0.9603

Note: GLCM=Gray Level Co-occurrence Matrix; ID=Inverse Difference; IDM=Inverse Difference Moment; IDMN= Inverse Difference Moment Normalized; IDN=Inverse Difference Normalized; IMC=Informational Measure of Correlation; MCC=Maximal Correlation Coefficient; GLSZM=Gray Level Size Zone Matrix; Features in bold denote selectee features.

  1. It was demonstrated that RFs could be correlated with others clinical parameters; since authors did not performed any correlation analysis, it cannot be excluded that radiomics score was a surrogate of other clinical variables.

>> Thank you for your comment. Yes, the radiomics features could be correlated with other important clinical variables. Following your comment, we correlated six important radiomics features common in all five folds with the important clinical variables that were available before surgery (i.e., tumor size, nodule-in-nodule appearance, rim calcification, and echogenicity) . The correlation analyses showed that the selected radiomics features had either weak correlation (r < 0.2) or high p-value (p > 0.1) (Supplementary Table S2). This confirmed that the selected radiomics features were not surrogates of other important clinical variables. The Results section was revised.

Supplemenatary Table S2. Correlation between selected radiomics features and important clinical variables. The first value in each cell element is r-value followed by p-value in the format of r-value (p-value).

Variables

Tumor Size

Echogenicity

Rim Calcification

Nodule-in-nodule appearance

Minimum

-0.034(0.661)

0.661(0.602)

-0.008(0.923)

0.072(0.351)

Elongation

0.055(0.474)

0.474(0.682)

0.010(0.899)

-0.016(0.833)

Sphericity

0.178(0.021)

0.021(0.514)

0.124(0.109)

-0.012(0.880)

Gray level non-uniformity normalized

-0.022(0.780)

0.780(0.987)

-0.036(0.641)

0.010(0.901)

Size zone nonuniformity

0.070(0.367)

0.367(0.376)

-0.025(0.747)

-0.047(0.541)

Small area low gray-level emphasis

-0.071(0.359)

0.359(0.962)

-0.003(0.969)

-0.069(0.372)

  1. The use of SVM classifier for this sample size is not justified and the risk of overfitting is very high.

>> Thank you for your comment. Since our study is a single-center one, we adopted the five-fold cross-validation strictly separating training and test data to reduce the risk of overfitting. We used a different set of features (on average 10 features across different folds) from 169 samples (134 FTC without metastasis and 35 FTC with metastatic cases) for the classifier. Many machine learning studies applied the SVM classifier where the minority class had less than 30 samples for 10 or more features [3-5]. Thus, we believe SVM classifier could be applied in our study. A theoretic study pointed out that if the features followed multivariate Gaussian distribution, the number of samples per class to apply SVM effectively should be greater than three times the number of features [6]. In our case, the cutoff was 30 (=10x3) on average due to the different number of selected features in the cross-validation. Thus, we have a theoretic rationale to apply SVM to our samples. Texts in the Methods and Discussions sections were revised.

  1. The authors did not provide any information about the calculated radiomics score as well as the full details of predictive model, and this does not allow to replicate the study.

>> Sorry for the omission. We provided the full details on how our radiomics score was computed in this revision. The SVM classifier with a linear kernel was constructed using the selected features. The parameters of SVM were weights that determined the decision hyperplane to separate two groups (i.e., FTC with and without metastasis). A signed distance was computed from features from the given sample to the hyperplane, which was further transformed using the sigmoid function to yield a probability value. The output probability value was assigned as the radiomics signature. The whole procedure was performed with a MATLAB command “fticsvm” with the prior uniform option. Texts in the Methods section were revised. Our computer code used in this study including the classifier was provided in the Supplementary material for future replication studies.

Specific Comment:

Line 74: Why authors decided to reduce the enrollment time for patients without distant metastases?

>> Thank you for your comment. The number of FTCs with distant metastasis was much smaller than FTCs without distant metastasis in our institution. As you know, planning case-control studies are usually advised to include no more than four or five controls per case because little statistical power is gained by further increasing this ratio. Therefore, we limited the enrollment period of FTCs without distant metastasis to five years to obtain statistical power.

Limitation was updated. “the enrollment period was not the same for patients with and without metastases. In fact, the number of FTCs with distant metastasis was much smaller than FTCs without distant metastasis in our institutional database. Because no more than four or five controls per case were recommend for case-control study to maintain statistical power, we limited the enrollment period of FTCs without distant metastasis to five years, while FTCs with distant metastasis had longer enrollment period.”

Line 88: Authors should compare the radiomics features extracted from images acquired by the different operators.

>> It is shown in the response above (Response #2).

Line 148: Usually univariate logistic regression is performed before the multivariate one: why did the authors do a correlation analysis?

Univariate analysis gives you a simplified marginal description between predictors and dependent variables and it is often used right before the multivariate analysis. The multivariate analysis gives you the complete picture of the variables and thus is more important than the results of univariate analysis. There are many choices in univariate analysis and we chose variants of correlation for the univariate analysis depending on the type of predictors. As you pointed out, since we adopted logistic regression as the multivariate analysis, the natural univariate analysis would be univariate logistic regression. Still, all univariate analysis evaluates a simple relationship between predictors and dependent variables and thus are similar. Following your comments, we performed univariate analysis using univariate logistic regression and confirmed that the significant results were the same in Table 4.

Line 186: Is the radiomics score a linear combination of the six LASSO selected features? Is it statistically different between the two groups of patients?

>> As shown in the response above (Response #5), the radiomics signature is a probability output of the SVM classifier using the selected features. The radiomics score was designed to discriminate between two groups (i.e., FTC with and without metastasis), and thus there should be a statistically significant difference in radiomics signature between two groups. This was confirmed with a two-sample t-test with a p-value less than 0.0001. Results section was updated.

Line 203: Authors should report the ICC results transparently.

>> As shown in the response above (Response #2), we provided full details of the ICC.

Line 205: Usually, the model parameters has to be reported in order to make the analysis repeatable.

>> As shown in the response above (Response #3), we provided full details of the model parameters.

Line 254: The authors stated that only 6/169 patients were acquired with an old ultrasound system: why did they not exclude these patients from the analysis?

>> Thank you for your question. Six were obtained from the old US system, but there was no problem in image analysis and the image quality was not bad. Rather, exclusion of these patients could induce selection bias.

Supplement X

SVM code

The code is available at https://github.com/skkuej/thyroid_SVM/blob/master/lasso_svm.mat.

%% data load, cross validation, z-normalization

X= data(:,3:62); % radiomics normalized_features(n=60),

y= data(:,2); % binary-metastases(n=169)

normalized_features = zscore(X(:,1:60));

foldMax = 5;

cvNum = 1;

c = cvpartition(y,'kfold',foldMax);

while cvNum <= foldMax

   trainingFeature = normalized_features(c.training(cvNum),:);

   testFeature = normalized_features(c.test(cvNum),:);

   trainingLabel = y(c.training(cvNum));

   testLabel = y(c.test(cvNum));

   %% Feature selection - Lasso

   lasso = cvglmnet(trainingFeature, trainingLabel, 'binomial');

   s(cvNum).selected_features = find(lasso.glmnet_fit.beta(:,(lasso.lambda == lasso.lambda_min)));

   trainingFeature = trainingFeature(:,s(cvNum).selectednormalized_features);

   testFeature = testFeature(:,s(cvNum).selectednormalized_features);

   %% SVM

   svmMdl = fitcsvm(trainingFeature,trainingLabel,'Prior','uniform');

   [labelHatTr, scoreTr] = svmMdl.predict(trainingFeature);

   [labelHatTs, scoreTs] = svmMdl.predict(testFeature);

   radiomics_score(cvNum).score = scoreTs;

   %% model evaluation

   [Xtr,Ytr,Ttr,AUCtr(cvNum)] = perfcurve(trainingLabel,scoreTr(:,2),1);

   [Xts,Yts,Tts,AUCts(cvNum)] = perfcurve(testLabel,scoreTs(:,2),1);

   conMat_train = confusionmat(trainingLabel, labelHatTr);

   ACC_train(cvNum) = (conMat_train(1,1)+conMat_train(2,2))/sum(conMat_train(:));

   SENS_train(cvNum) = conMat_train(2,2)/(conMat_train(2,1)+conMat_train(2,2));

   SPEC_train(cvNum) = conMat_train(1,1)/(conMat_train(1,1)+conMat_train(1,2));

   conMat_test = confusionmat(testLabel, labelHatTs);

   ACC_test(cvNum) = (conMat_test(1,1)+conMat_test(2,2))/sum(conMat_test(:));

   SENS_test(cvNum) = conMat_test(2,2)/(conMat_test(2,1)+conMat_test(2,2));

   SPEC_test(cvNum) = conMat_test(1,1)/(conMat_test(1,1)+conMat_test(1,2));

   cvNum = cvNum + 1;

end

Round 2

Reviewer 2 Report

Authors have answered to all my questions and made the necessary changes to the manuscript.